# CAtNIPP: Context-Aware Attention-based Network for Informative Path Planning

**Yuhong Cao**
National University of Singapore
caoyuhong@u.nus.edu

**Yizhuo Wang**
National University of Singapore
wy98@u.nus.edu

**Apoorva Vashisth**
Indian Institute of Technology, Kharagpur
av1999@iitkgp.ac.in

**Haolin Fan**
National University of Singapore
e0816265@u.nus.edu

**Guillaume Sartoretti**
National University of Singapore
mpegas@nus.edu.sg

**Abstract:**

Informative path planning (IPP) is an NP-hard problem, which aims at planning a path allowing an agent to build an accurate belief about a quantity of interest throughout a given search domain, within constraints on resource budget (e.g., path length for robots with limited battery life). IPP requires frequent online replanning as this belief is updated with every new measurement (i.e., *adaptive* IPP), while balancing short-term exploitation and longer-term exploration to avoid suboptimal, *myopic* behaviors. Encouraged by the recent developments in deep reinforcement learning, we introduce CAtNIPP, a fully reactive, neural approach to the adaptive IPP problem. CAtNIPP relies on self-attention for its powerful ability to capture dependencies in data at multiple spatial scales. Specifically, our agent learns to form a *context* of its belief over the entire domain, which it uses to sequence local movement decisions that optimize short- and longer-term search objectives. We experimentally demonstrate that CAtNIPP significantly outperforms state-of-the-art non-learning IPP solvers in terms of solution quality and computing time once trained, and present experimental results on hardware.

**Keywords:** deep RL, informative path planning, context-aware decision-making

## 1 Introduction

In many real-life robotic deployments that involve data acquisition, such as mapping/exploration of unknown areas for inspection or search-and-rescue applications, environmental monitoring, and surface inspection/reconstruction [1, 2, 3], an autonomous robot needs to plan a path to visit a given domain and obtain measurements about a scalar field of interest, without *a priori* knowledge of the true underlying distribution of this information. That is, starting from a uniform distribution with high uncertainty, the agent must construct a belief over the distribution of *interest* throughout the domain (e.g., target likelihood, temperature, surface roughness) based on successive measurements along its path. This problem is known as the *informative path planning* (IPP) problem. Specifically, IPP aims to plan a path that maximizes *information gain*, while satisfying a budget constraint (e.g., path length for robots with limited battery life). IPP problems can be further classified as either *non-adaptive* or *adaptive*. Non-adaptive solvers pre-plan a complete path offline and execute this pre-determined path, without any replanning upon obtaining new measurements online [4, 5, 6]. On the other hand, adaptive solvers replan the search path frequently as the agent's belief is updated based on new measurements [1, 2, 7]. While our approach can also be used for non-adaptive IPP, we focus on the more general adaptive IPP problem for its wider applicability to real-life robotic tasks.

Differently from general path planning problems, where the agent is often assigned a goal position, IPP requires the agent to identify and visit all potential interesting areas throughout the environment.

6th Conference on Robot Learning (CoRL 2022), Auckland, New Zealand.

Therefore, efficient IPP solvers must reason about the entire agent's belief to make non-myopic decisions [8], which balance short-term exploitation of known interesting areas with longer-term exploration of unknown areas in the domain. Many IPP solvers rely on computationally expensive means to optimize long-horizon trajectories [4, 5, 7]. Trading off solution quality in favor of lower computing times, more recent approaches have embraced sampling-based planning [7, 9, 10].

To further improve computing time and solution quality, we introduce CAtNIPP, a deep reinforcement learning (dRL) based framework for 2D adaptive IPP. We first decrease the complexity of our continuous-space search domain by generating a *probabilistic roadmap* [11], i.e., a random sparse graph that covers the domain. We then associate this roadmap with the agent's belief, and formulate adaptive IPP as a sequential decision-making problem on this graph. We propose

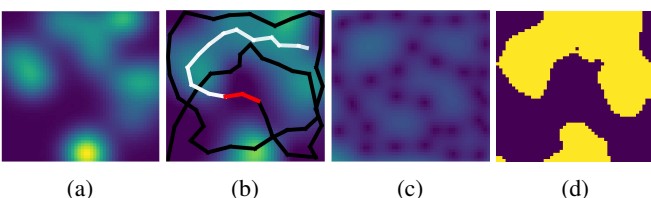

|  (a)  |  (b)  |  (c)  |  (d)  |

Figure 1: **CAtNIPP's *trajectory sampling* variant for adaptive IPP**, showing the path executed by the agent so far (black), and the long-horizon trajectory that was just (re-)planned (white), of which the red portion will be executed until the next replanning step. a) True interest map (unknown to the agent), b) Predicted interest map from all measurements so far, c) Associated predicted standard deviation, and d) Predicted high-interest areas.

and train an attention-based neural network that outputs a policy to select which neighboring node to visit next, thus allowing us to iteratively plan an (adaptive) informative path. There, self-attention over the graph nodes allows the agent to construct a global *context*, by embedding its entire belief into local decision features while identifying dependencies between nodes/areas at different spatial scales. In doing so, the neural-based, reactive nature of our approach drastically improves planning time, compared to planners that optimize full trajectories, while context-awareness helps improve solution quality by allowing our agent to identify and sequence *non-myopic* decisions that near-optimally balance short- and long-term objectives. The main contributions of this work are:

- We propose a new fully reactive, policy-based dRL framework for adaptive IPP, which significantly outperforms state-of-the-art IPP solvers in terms of both solution quality and computing time. In CAtNIPP, the agent learns a subtle global representation of its entire belief over the domain, that allows it to sequence non-myopic decisions that can achieve longer-term objectives.

- We further propose to rely on receding-horizon, sampling-based trajectory optimization (Fig. 1) to output higher-quality, longer-horizon trajectories by leveraging the nature of our stochastic policies, while remaining tractable and usable in real time. We demonstrate these variants of CAtNIPP in simulation as well as on hardware on a light-intensity-based IPP task.

## 2 Related Work

To reduce the computing time and improve solution quality, most recent IPP methods have relied on randomized, sampling-based methods. For non-adaptive IPP solvers, Hollinger *et al.* [9] introduced the *RIG-tree* algorithm, which utilizes RRT-star (a Rapidly-exploring Random Tree variant [12]) to randomly build a tree structure used to explore the environment and maximize information gain. Arora *et al.* [5] combined Constraint Satisfaction and Travelling Salesman Problems to introduce *Randomized Anytime Orienteering* (RAOr). RAOr iteratively samples the search space and solves a TSP instance to visit all sampled locations within the given budget constraints. According to Popović *et al.* [1], despite RIG-tree and RAOr not having been initially designed for adaptive IPP, they only require seconds to plan, which allows them to be generalized to adaptive online replanning scheme in a traditional receding-horizon manner. Regarding solvers initially designed for adaptive IPP, Hitz *et al.* [7] proposed an evolutionary strategy to achieve state-of-the-art performance. They applied CMA-ES to generate candidate solutions from a multi-variate Gaussian distribution, where the mean and covariance matrices are adaptively updated according to the evaluation of the candidates. Apart from these conventional solvers, there has been a couple of recent, value-based RL solvers for IPP. Wei *et al.* [6] proposed an RNN-based solver, which reasons about the positions of measurement (for their predicted reduction in uncertainty) but without considering measurement values, thus limiting its use to non-adaptive IPP problems. Ruckin *et al.* [13] proposed a specialized CNN-based solver for 3D IPP, developed for a specific problem statement that assumes image-like measurements and relies on Kalman Filtering for belief update.

## 3 Background

In this section, we first introduce Gaussian Processes (GPs), which are used to model the agent's belief (i.e., predicted interest map). We then formulate the general definition of our IPP problem based on such a GP. Finally, we describe the adaptive replanning requirement for adaptive IPP.

**Gaussian Process** In IPP, *interest* (e.g., target distribution, temperature, or radiation level), is associated with the 2D environment $\mathcal{E} \subset \mathbb{R}^2$ and modeled as a continuous function $\zeta : \mathcal{E} \to \mathbb{R}$. Gaussian Processes have been widely used to represent such a continuous interest distribution, by providing a natural means to interpolate between discrete measurements [1, 6, 7], so that $\zeta \approx \mathcal{GP}(\mu, P)$. Specifically, given a set of $n'$ locations $\mathcal{X}^* \subset \mathcal{E}$ at which interest is to be inferred, a set of $n$ observed locations $\mathcal{X} \subset \mathcal{E}$ and the corresponding measurements set $\mathcal{Y}$, the mean and covariance of the GP are regressed as: $\mu = \mu(\mathcal{X}^*) + K(\mathcal{X}^*, \mathcal{X})[K(\mathcal{X}, \mathcal{X}) + \sigma_n^2 I]^{-1}(\mathcal{Y} - \mu(\mathcal{X})), P = K(\mathcal{X}^*, \mathcal{X}^*) - K(\mathcal{X}^*, \mathcal{X})[K(\mathcal{X}, \mathcal{X}) + \sigma_n^2 I]^{-1} \times K(\mathcal{X}^*, \mathcal{X})^T$, where $K(\cdot)$ is a pre-trained/selected kernel function, $\sigma_n^2$ is a hyperparameter describing the measurement noise, and $I$ is the $n \times n$ identity matrix. In this work, following [1, 10], we use the Matérn $3/2$ kernel function.

**Informative Path Planning** The general IPP problem aims to find an optimal trajectory $\psi^*$ in the space of all available trajectories $\Psi$ for maximum gain in some information-theoretic measures:

$$\psi^* = \underset{\psi \in \Psi}{\arg\max} \, \mathrm{I}(\psi), \text{ s.t. } \mathrm{C}(\psi) \leq B, \tag{1}$$

where $\mathrm{I} : \psi \to \mathbb{R}^+$ is the information gained from the measurements obtained along the trajectory $\psi$ , $\mathrm{C} : \psi \to \mathbb{R}^+$ maps a trajectory $\psi$ to its associated execution cost, and $B \in \mathbb{R}^+$ is the given path-length budget. Following [5, 6, 7], the trajectory $\psi$ is given a start and a destination but we note that our method can be easily extended to remove the need for a destination. To evaluate the information gained from measurements, following [1, 4], we use the variance reduction of the GP to represent information gain: $\mathrm{I}(\psi) = \mathrm{Tr}(P^-) - \mathrm{Tr}(P^+)$, where $\mathrm{Tr}(\cdot)$ denotes the trace of a matrix, $P^-$ and $P^+$ are the prior and posterior covariances, which are obtained before and after taking measurements along the trajectory $\psi$. In this work, to model the data collection of common sensors, we let the agent take a measurement every time it has traveled a fixed distance from the previous measurement, thus the number of measurement is only determined by the path length budget $B$.

**Adaptive Replanning** If the information gain only depends on the covariance $P$, i.e., the location of measurements, the objective is considered *non-adaptive* since the trajectory could be entirely planned offline ahead of time, based on the agent's initial (often uniform) belief. However, in real-world applications such as search-and-rescue, we usually aim to discover regions of high interest and further cover (exploit) them. To this end, following [1, 7], we rely on the *upper confidence bound* to define high-interest areas $\mathcal{X}_I$: $\mathcal{X}_I = \{x_i \in \mathcal{X}^* | \mu_i^- + \beta P_{i,i}^- \geq \mu_{th}\}$, where $\mu_i^-$ and $P_{i,i}^-$ are the prior mean and variance of the GP at the measurement location $x_i$. $\mu_{th}$ and $\beta \in \mathbb{R}^+$ are used to control the threshold and confidence interval respectively ($\mu_{th} = 0.4, \beta = 1$, in practice). By replacing $\mathcal{X}^*$ with $\mathcal{X}_I$ in the covariance calculation, we restrict the information gain in the objective function Eq. (1) to the high-value areas predicted by the GP. This formulation makes the IPP objective dependent on the measurement values in addition to their location, making the problem truly *adaptive*. Therefore, frequent online replanning of the trajectory is required to minimize uncertainty in the (now dynamically defined) high-interest areas $\mathcal{X}_I$.

## 4 Method

In this section, we cast adaptive IPP as an RL problem and detail our attention-based neural network, as well as our long-horizon planning strategy to further boost the performance of a learned policy.

### 4.1 IPP as a RL Problem

**Sequential Decision-making Problem** First, to avoid the complexity associated with a continuous search domain, we rely on probabilistic roadmaps (PRM) [11] to build a route graph $G = (V, E)$, with $V$ a set of uniformly-random-sampled nodes over the domain, and $E$ a set of edges. Each node $v_i = (x_i, y_i) \in V$ is connected to its $k$ nearest neighboring nodes and $v_0$ is the destination. Then, to solve the adaptive IPP using RL, we formulate it as a sequential decision-making problem on this graph. That is, we let agent interact with the environment by choosing which node to move to from

amongst the neighbors of its current node. Movement between nodes happens as a straight line. As a result, the agent's trajectory $\psi$ can be represented as an ordered set $(\psi_s, \psi_1, ..., \psi_d), \forall \psi_i \in V$, where $\psi_s$ and $\psi_d$ denote the start and destination nodes respectively. As a result, the trajectory is adaptively planned, since it is constructed from sequential movements, each depending on the agent's global belief, which gets updated online based on new measurements.

**Observation** The observation $s_t = \{G', v_c, B_c, \psi_{s,c}, M\}$ of our IPP agent consists of three parts: the augmented graph, the planning state, and the budget mask.

The augmented graph $G' = (V', E)$ is used to describe the environment modeled by the GP. It is a combination of the route graph $G = (V, E)$ and the GP $\mathcal{GP}(\mu, P)$, where each node $v'_i = (v_i, \mu(v_i), P(v_i)) \in V'$. This augmented graph stores the information about the agent's global belief and determines the agent's local action space. The planning state is defined by $\{v_c, B_c, \psi_{s,c}\}$, where $v_c \in V$ is the current position of the agent, $B_c = B - \mathrm{C}(\psi_{s,c})$ is the remaining budget, and $\psi_{s,c} = (\psi_s, \psi_1, ..., v_c)$ is the executed trajectory so far. The budget mask $M$ is a binary vector containing one element for each node in the route graph, stating whether selecting this node at the current step would result in violating the budget constraint. To obtain this mask, we pre-solve the shortest path problem using Dijkstra [14] to compute the minimal cost to the current node and to the destination from each node $v_i$. We then compute a virtual budget $B^*_i = B_c - \min \mathrm{C}(v_c, v_i) - \min \mathrm{C}(v_i, \psi_d)$ for each node based on the planning state. Finally, according to $B^*$, we compute each entry of $M$ as $M_i = \begin{cases} 1 & \text{if } B^*_i < 0 \\ 0 & \text{otherwise.} \end{cases}$ That is, actions (i.e., neighboring nodes) that would inevitably result in budget overruns are specifically filtered out by the budget mask $M_i$, which iteratively guarantees the completeness of CAtNIPP by forcing the agent towards the destination via the shortest possible path when the budget is about to run out.

**Action** Each time the agent reaches a node, the GP is updated based on all measurements obtained so far, and the agent immediately selects its next action. Specifically, at each such decision step $t$, given the agent's observation, our attention-based neural network outputs a stochastic policy to select the next node to visit out of all neighboring nodes. The policy is parameterized by the set of weights $\theta$: $\pi_\theta(\psi_t = v_i, (v_c, v_i) \in E \mid s_t)$, where $E$ is the edge set of the underlying graph.

**Reward** At each decision step, to maximize information gain, the agent is given a positive reward based on the reduction in uncertainty associated with its most recent action: $r_t = (\mathrm{Tr}(P^{t-1}) - \mathrm{Tr}(P^t))/\mathrm{Tr}(P^{t-1})$, where we experimentally found that scaling the reward by $\mathrm{Tr}(P^{t-1})$ helped stabilize training by keeping the rewards consistent in magnitude. However, this normalization introduces a deviation between the training objective and the IPP objective. Therefore, at the last decision step of each episode, we introduce a negative correction reward $r_d = -\alpha \cdot \mathrm{Tr}(P^d)$, where $P^d$ is the covariance after executing the whole trajectory $\psi$, and $\alpha$ is a scaling factor (1 in practice). We empirically observed that this correction reward helps mitigate the bias introduced by the (dynamic) normalization used in our dense rewards, towards the true IPP reward.

## 4.2 Neural Network Structure

The proposed attention-based neural network consists of an encoder and a decoder modules (see Fig. 2). We use the encoder to model the observed environment by learning the dependencies between nodes in the augmented graph $G'$, i.e., the *context*. Based on the features extracted by the encoder, the planning state $\{v_c, B_c, \psi_{s,c}\}$, and the budget mask $M$, the decoder then outputs the policy over which neighboring node to visit next. To handle graphs with arbitrary topologies, our encoder uses a standard Transformer attention layer with graph Positional Encoding (PE) based on the graph Laplacian's eigenvector [15], thus providing the neural network with the ability to reason about node connectivity. While general policy-based RL agents have a fixed action space, our decoder is inspired by the Pointer Network [16] to allow the dimension of the final policy to depend on the number of neighboring nodes, allowing our network to generalize to arbitrary graphs.

**Attention Layer** The Transformer attention layer [17] is used as the fundamental building block in our model. The input of such an attention layer consists of the query source $h^q$ and the key-and-value source $h^{k,v}$. The attention layer updates the query source using the weighted sum of the value vector, where the attention weight depends on the similarity between key and query. We compute the updated feature $h'_i$ as: $q_i = W^Q h^q_i$, $k_i = W^K h^{k,v}_i$, $v_i = W^V h^{k,v}_i$, $u_{ij} = \frac{q_i^T \cdot k_j}{\sqrt{d}}$, $a_{ij} = \frac{e^{u_{ij}}}{\sum_{j=1}^n e^{u_{ij}}}$, $h'_i = \sum_{j=1}^n a_{ij} v_j$, where $W^Q, W^K, W^V$ are $d \times d$ learnable matrices. The updated fea-

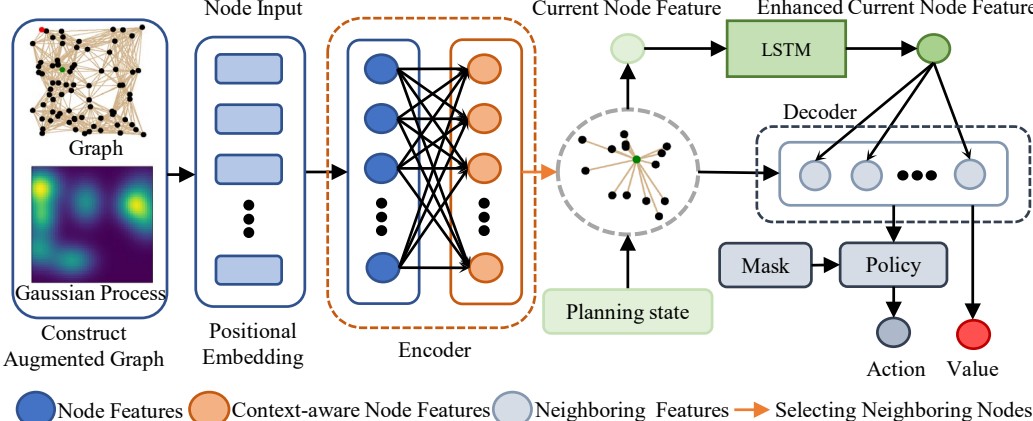

**Figure 2: CAtNIPP's attention-based neural network.** The encoder module relies on self-attention to identify and represent the global dependencies between nodes in the agent's belief (i.e., augmented graph) as *context-aware node features*. Relying on the current and neighboring context-aware node features (grey dashed circle), the planning state, and the mask, the decoder relies on cross-attention to output the final, context-aware policy (and value estimate during training).

tures are then passed through the feed forward sublayer, composed of two linear layers and a ReLU activation. As in [17], we use layer normalization and residual connections in these two sublayers.

**Encoder** The encoder is used to model the observed environment by learning dependencies between nodes in the augmented graph $G'$. We first embed the *node inputs* $V'$ into $d$-dimensional *node features* $h_i^n$ and add Laplacian positional embeddings: $h_i^n = \begin{cases} W^L v_i' + b^L + W^{PE}\lambda_i + b^{PE} & i > 0 \\ W^D v_0' + b^D + W^{PE}\lambda_0 + b^{PE} & i = 0 \end{cases}$ where $\lambda_i$ is the pre-computed $k$-dimensional Laplacian eigenvector, and $W^L, W^D \in \mathbb{R}^{d \times 4}$, $W^{PE} \in \mathbb{R}^{d \times k}$, $b^L, b^D, b^{PE} \in \mathbb{R}^d$ are learnable parameters. Note that the destination node $v_0'$ is embedded by another linear layer. The node features are then passed to an attention layer, where $h^q = h^{k,v} = h^n$, as is commonly done in self-attention mechanisms. We term the output of the encoder, $h^{en}$, the *context-aware node features*, since each of these updated node features $h_i^{en}$ contains the dependencies of $v_i'$ with all other nodes.

**Decoder** The decoder is used to output a policy based on the context-aware node features, the planning state $\{v_c, B_c, \psi_{s,c}\}$, and the budget mask $M$. We first merge the information about the budget and the high-interest areas to the context-aware node features: $\hat{h}_i^{en} = W^B[h_i^{en}, B_i^*, \mu_{th}] + b^B$, where $W^B \in \mathbb{R}^{d \times (d+2)}$ and $b^B \in \mathbb{R}^d$ are learnable parameters. Then, according to the current position $v_c$ and the edge set $E$, we select the *current node feature* $h_c^{en}$, the *neighboring features* $h_n^{en}$ from $\hat{h}^{en}$, and the *neighbor mask* $M_n$ from $M$. After that, the current node feature $h_c^{en}$ are passed to a LSTM block, where the hidden state and cell state are input from previous current node feature along the executed trajectory $\psi_{s,c}$. The LSTM output $\hat{h}_c^{en}$ is merged with the *destination feature* $\hat{h}_0^{en}$ to compute the *enhanced current node feature* $h^{ec}$: $h^{ec} = W^C[\hat{h}_c^{en}, \hat{h}_0^{en}] + b^C$, where $W^C \in \mathbb{R}^{d \times 2d}$ and $b^C \in \mathbb{R}^d$ are learnable parameters. We feed the enhanced node current feature and the neighboring features to an attention layer, where $h^q = h^{ec}$ and $h^{k,v} = h^n$. We denote the output of this attention layer $\hat{h}^{ec}$, which is simultaneously passed to a linear layer to output the state value $V(s_t)$, and to the final attention layer with the neighboring features, where $h^q = \hat{h}^{ec}$ and $h^{k,v} = h^n$. For this final attention layer, we directly treat the attention weights $a_i$ as the final policy $u_i$ for the IPP agent, where invalid nodes are explicitly masked using $M^n$. The masked $u_i$ is finally normalized to yield the probability distribution $\pi$ for the next node to visit: $\pi_i = \pi_\theta(\psi_t = v_i | s_t) = e^{u_i} / \sum_{i=1}^n e^{u_i}$.

### 4.3 Training

Our model is trained using PPO [18]. At the beginning of each training episode, we average 8 to 12 random 2-dimensional Gaussian distributions in the unit square $[0, 1]^2$, to construct the true interest map. The robot's belief starts as a uniform distribution $\mathcal{GP}(0, P^0)$, $P_{i,i}^0 = 1$. The start and destination positions are randomly generated in $[0, 1]^2$. During training, the number of nodes for our

Table 1: **Comparison with SOTA IPP solvers (10 trials on 30 instances for each budget).** Tr(P) is the average covariance matrix trace after running out of budget (standard deviation in parentheses). T(s) is the average total planning time in seconds.

| Method | Budget 6 | | Budget 8 | | Budget 10 | | Budget 12 | |
|---|---|---|---|---|---|---|---|---|
| | Tr(P) | T(s) | Tr(P) | T(s) | Tr(P) | T(s) | Tr(P) | T(s) |
| RIG-Tree | 32.69($\pm$12.86) | 132.36 | 15.44($\pm$5.49) | 192.74 | 7.74($\pm$3.01) | 240.58 | 4.80($\pm$2.21) | 291.31 |
| RAOr | 26.80($\pm$15.16) | 17.12 | 11.17($\pm$3.87) | 40.13 | 6.28($\pm$2.25) | 73.60 | 4.71($\pm$1.13) | 127.44 |
| CMA-ES | **17.44**($\pm$6.09) | 124.23 | 10.48($\pm$5.38) | 181.41 | 6.77($\pm$3.74) | 241.47 | 4.51($\pm$2.42) | 268.69 |
| g.(800) | 22.86($\pm$6.42) | **1.23** | 7.72($\pm$2.77) | **1.68** | 3.97($\pm$1.46) | **2.20** | 2.70($\pm$1.18) | **2.52** |
| ts.(4) | 20.19($\pm$3.88) | 90.31 | **7.04**($\pm$1.44) | 123.56 | **3.82**($\pm$0.61) | 158.44 | **2.52**($\pm$0.41) | 194.97 |

graph is randomized within $[200, 400]$ for each episode, the number of neighboring nodes is fixed to $k = 20$, and the budget is randomized within $[6, 8]$. A measurement is obtained every time the agent has traveled $0.2$ from the previous measurement. We set the max episode length to $256$ time steps, and the batch size to $1024$. We use the Adam optimizer with learning rate $10^{-4}$, which decays every $32$ steps by a factor of $0.96$. For each training episode, PPO runs 8 iterations. Our model is trained on a workstation equipped with a i9-10980XE CPU and four NVIDIA RTX 3090 GPUs. We train our model utilizing Ray, a distributed framework for machine learning [19]. We run 32 IPP instances in parallel to accelerate the data collection and training, and need around 24h to converge.[12]

### 4.4 Trajectory Sampling

Until now, we discussed solving the IPP in the standard RL manner, i.e., iteratively selecting the next node to visit each time the agent reaches a node. Inspired by conventional non-learning IPP solvers, we further propose a receding-horizon strategy for our RL agent, where an $m$-step trajectory is output at each (re)planning step but only a portion of it is executed before the next replanning step (see Fig. 1). We utilize the learned policy for further optimization by *sampling*, which has been shown to be a reliable optimization strategy for learning-based routing planner [20, 21]. That is, at each planning step, based on the learned policy, our *trajectory sampling* method parallely plans a number $s$ of $m$-step trajectories, and then selects the trajectory that maximizes the information gain as the final trajectory $\psi^*$. During the $m$-step planning process, only the covariance of the GP can be predicted, since no measurement is actually taken before executing the trajectory.

## 5 Experiments

In this section, we compare CAtNIPP with state-of-the-art (SOTA) baselines IPP solvers on a fixed set of randomly generated environments with identical randomized conditions. We also present numerical and experimental validation of CAtNIPP on an light-intensity-based adaptive IPP task. In our supplemental material, we also tested a number of variants of our model and its generalizability.

### 5.1 Comparison Results

We compare CAtNIPP against a number of state-of-the-art IPP solvers: (a) CMA-ES [7] (we use linear B-spline for the CMA-ES solver for a fair comparison), (b) RAOr [5], and (c) RIG-tree [9]. Following [1], we implement RAOr and RIG-tree in a receding-horizon manner to make them adaptive. All considered solvers (except our fully reactive, greedy variants) replan paths after executing $0.4$ of their previously planned trajectory. Starting from hyperparameters suggested by their original papers, we tuned these solvers to output highest-quality solutions, while keeping the total planning time similar to our trajectory sampling variants. This enables us to offer a fair comparison between our methods and these baselines. CAtNIPP's *greedy* and *trajectory sampling* variant are denoted by g.($n$) and ts.($m$) respectively, where $n$ is the number of nodes for the route graph and $m$ is the number of trajectories sampled at each planning step ($n$ is fixed to $400$ for ts.). Greedy variants work in the standard RL manner, i.e., the agent always selects the action with highest activation in its policy. Trajectory sampling variants plan a 15-step trajectory and execute the first 3 steps before replanning (in practice, $\sim 0.4$ of traveled distance), following the receding-horizon setup in [7].

---

[1]Actually, 12 hours of training on one NVIDIA RTX 2080 (batch size 256) can yield similar performance.

[2]Our full code and trained model can be found at: `https://github.com/marmotlab/CAtNIPP`

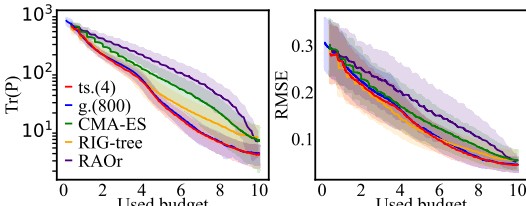

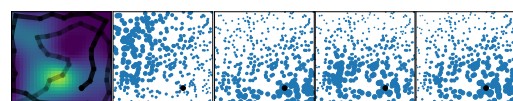

Figure 3: **Comparison with SOTA IPP solvers for a fixed budget of 10 (10 trials on 30 instances).** Our two CAtNIPP variants significantly outperform all solvers in reducing uncertainty (left) as well as root mean square error (right).

Figure 4: **Attention weights visualization from the trained encoder.** The query source is the node at the current position (black) and the keys source are nodes in the augmented graph (blue). One attention head and three attention heads pay attention to longer- and shorter-term high-interest areas respectively.

We report the uncertainty remaining (covariance matrix trace $\mathrm{Tr}(P)$ in the high-interest areas $\mathcal{X}_I$, lower is better) after finishing the mission, as well as the total planning time in Table 1. The evolution of the remaining uncertainty and root mean square error compared to the ground truth $||\mu - \zeta||_2$ are plotted in Fig. 3. Upon closer inspection, we note that RAOr tends to plan trajectories that exploit nearby high-interest areas, thus reducing the uncertainty slowly in the early stage and more rapidly later on. RIG-tree, on the other hand, has a strong tendency for exploration, which leads to a fast uncertainty reduction earlier on, but a slower one in the later stages of an episode. As a meta-heuristic solver, CMA-ES finds a good trade-off between exploration and exploitation, resulting in the best overall performance among non-learning solvers (even better than CAtNIPP with budget 6).

Fig. 3 shows that CAtNIPP maintains best overall performance with respect to all metrics throughout the whole budget span. We performed paired t-tests between the covariance reduction results of our variants and each of the baselines (6 tests in total), which all yielded p-values lower than $1.06 \cdot 10^{-4}$. Using a Bonferroni correction for these multiple tests, we find the final significance threshold $p = 1.67 \cdot 10^{-3}$ (for a standard, original $p = 0.01$ threshold), indicating that CAtNIPP (most likely) significantly outperforms all other baselines in terms of covariance reduction. We further note that our ts. variants exhibit improved solution quality over our greedy variants ($8\%$ better in average) by further refining these solutions, e.g., in some extreme cases where greedy variants do particularly poorly. However, greedy variants plan up to around $100\times$ faster than ts. variants and other SOTA IPP solvers, making it the most time-efficient CAtNIPP version.

## 5.2 Generalization Results

In our supplemental material, we include detailed testings of CAtNIPP's generalization capabilities. We demonstrate that our model, trained on randomly generated ground truths, can generalize to completely different environments never seen during training, such as different Gaussian mixture models or even handcrafted, non-Gaussian distributions, while still outperforming baselines. Our results suggest that CAtNIPP endows the agent with a general strategy for IPP, rather than overfitting to a particular class of ground truths.

## 5.3 Attention Visualization: Learning to Be Context-aware

We believe that the superior solution quality of CAtNIPP mainly comes from its ability to be *context-aware*, and thus to avoid the type of short-sightedness usually associated with local, reactive IPP planners. We investigated the learned attention mechanism at the core of CAtNIPP by visualizing learned attention weights (larger dots means higher weight) at representative time steps. In particular, Fig. 4 shows that the left attention head and the right three attention heads of the encoder have learned to focus on the longer- and shorter-term high-interest regions respectively. Given these context-aware node features, the agent finally learns to make local decisions that can optimize objectives at the different scales identified by the encoder. Our ablation results in the supplemental material further confirm that the presence of the encoder is critical.

## 5.4 Numerical and Experimental Validation

We carried out experiments to validate CAtNIPP's performance on a TurtleBot3 robot, over a printed grayscale image of $2.38 \times 2.38 \mathrm{m}^2$ representing the ground truth interest map (see Fig. 5). The robot

is equipped with an on-board camera used to measure the ground light (grayscale) intensity (as an example of a simple onboard intensity-level sensor, e.g., temperature/radioactivity/gas levels), while its position is obtained by a downward-facing, overhead camera.

In this experiment, a trained CAt-NIPP model adaptively outputs the next node location to visit based on the agent's current belief, using a 400-nodes graph, and the agent only takes measurements when reaching a node. This experiment confirms that CAtNIPP is easily deployable on robot for online, reactive planning, and highlights its low computational cost ($\leq 0.1$s per decision on CPU). Our supplemental material includes simulation videos in environments of up to $8 \times 8\text{m}^2$.

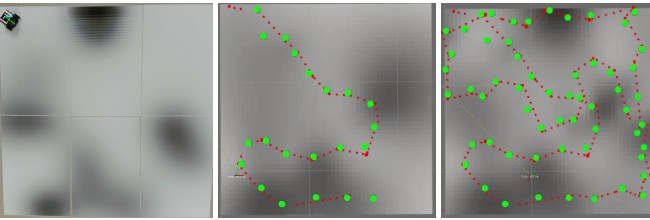

   (a) True interest map   (b) Early agent belief   (c) Final agent belief

Figure 5: **Experimental validation of CAtNIPP on Turtle-Bot3**. Selected nodes are shown in green, and robot trajectory in red (intermediate path in (b), and full final path in (c)).

## 6 Limitations

We believe that the limitations of CAtNIPP lie at three different levels: generalizability, area discretization, and implementation readiness:

- Our proposed method is sensitive to changes in the hyperparameters of the GP's kernel function and the upper bound confidence that defines high-interest areas. Currently, CAtNIPP would require retraining if any of these parameters were changed significantly. In practice, the hyperparameters of the GP's kernel are decided by key characteristics of the actual sensor equipped on the robot (e.g., range or resolution of a camera). Nevertheless, based on our experience, for a given type of real-world task (e.g., search and rescue, fire detection, agriculture monitoring), the definition of "high-interest" is usually rather static. Thus we would expect that, after an initial training based on these parameters, our learned model will likely not need any further training in practice, since retraining would only be needed if the task or onboard sensor are drastically changed.

- We currently assume uniform sampling of the route graph, which may prevent the agent from reaching an interesting area due to insufficient graph coverage. However, CAtNIPP is already able to handle arbitrary graphs. To address this issue, especially in later stages of the planning, we will further investigate online re-sampling of graph nodes, e.g., according to the current belief.

- We currently plan paths in a simplified graph, where paths between nodes are straight lines, ignoring most real-life robot motion constraints (i.e., holonomic robot assumption). Future work will explicitly consider the robot's motion model, e.g., in the state representation (velocity, heading, kinematic/dynamics constraints), to better trade-off robot-specificity with ease of implementation.

## 7 Conclusion

In this paper, we introduce CAtNIPP, a policy-gradient-based dRL method for adaptive IPP that relies on self-attention to endow the agent with the ability to sequence local decisions, informed by its global context over the search domain to avoid short-sightedness. In addition to solving adaptive IPP by simply greedily exploiting our learned policy, which can be done at very low computational cost ($\sim 0.1$s per decision once trained), we propose a sampling-based strategy that utilizes the learned policy more efficiently to output higher-quality solutions, while keeping the computing time on par with existing IPP solvers. We experimentally demonstrate that both variants of CAtNIPP significantly outperform state-of-the-art IPP solvers in terms of solution quality and planning time, with strong generalization to classes of distributions never seen during training. Finally, we present experimental results on physical and simulated robots in a representative online, adaptive IPP task, showing promises for robotic deployments in real-life monitoring, inspection, or mapping scenarios.

Future work will mainly focus on extending our model to multi-agent IPP, where robots need to reason about each other to cooperatively plan informative paths, by leveraging synergies and avoiding redundant work. We also plan to investigate the use of CAtNIPP for robot exploration tasks, where more real-world object such as obstacles and sensors need to be considered in the planning process.

## Acknowledgments

This work was supported by Temasek Laboratories (TL@NUS) under grant TL/SRP/21/19.

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
