# OpenReview forum: "CAtNIPP: Context-Aware Attention-based Network for Informative Path Planning"
_robot-learning.org/CoRL/2022/Conference — CoRL 2022 Poster_

### Official Review · Reviewer_HgVR · 2022-07-15

**Originality:** Very Good
**Technical Quality:** Very Good
**Clarity Of Presentation:** Excellent
**Impact:** 4

**Recommendation:**

Weak Accept: I recommend accepting the paper, but will not argue for my recommendation if the majority of other reviewers have a different opinion.

**Summary:**

This paper proposes a method for informative path planning (IPP). In IPP, the task is to design a trajectory for sampling locations of a Gaussian process with maximum reduction in uncertainty, while respecting a budget constraint.
The paper proposes to remap the IPP problem as a Markov decision process via a probabilistic roadmap graph, and applies reinforcement learning (RL) with neural network (NN) based function approximation to solve for a policy. Besides the remapping of the problem, the main innovations of the paper are the NN structure using attention/transformers and adapting it to handle arbitrary graph topologies.
The proposed method is compared to three IPP algorithms from the literature. The proposed method can perform very well, that the attention mechanism is crucial for the good performance, and that the method can also to some degree generalize beyond its training data. A proof-of-concept demonstration with real robot hardware is also given.

**Issues:**

Could you comment or on address the weaknessess listed above, and correct the minor issues?

**Quality Of The Limitations Section:**

Limitations are addressed clearly

**Reviewer Expertise:**

4: The reviewer is confident but not absolutely certain that the evaluation is correct

**Robotics Focus:**

Sufficient demonstration on hardware

**Strengths And Weaknesses:**

# Strengths

- The empirical performance compared to previous state-of-the-art is strong. The ablation studies also confirm that the attention mechanism proposed in the encoder is crucial for good performance, comparing the significance of the papers main contribution.

- The paper is technically sound, and clearly describes its methodology. The RL reformulation seems particularly useful, and the idea of discretizing the task using a probabilistic roadmap is quite nice. This could be of interest to practicioners in other tasks requiring discretization as well. It also seems like this approach could easily be extended to environments with more complex topologies.

- The section on limitations was very clear and comprehensive. Well done!

- The generalization analysis in the supplementary material is quite interesting, indicating that the proposed method can also work for field intensity distributions that strongly differ from the Gaussian mixtures used during training.

# Weaknesses

- My main concern is how applicable this method would be when the process to be identified is not that well known, or the ability to generate training data for the method is limited or cannot match the actual target process. The generalization study does give some positive indications, but a more thorough analysis and discussion of this would be good.

- I suppose the times in Table 1 do not include the training times for the proposed method. It seems that the comparison as presented is not entirely fair, and in particular RAOr seems a strong competitor.

- It was not clear what the plus/minus values in parentheses in Table 1 or the shading in Figure 3 correspond to. Depending on this, it is possible that the observed improvements are not statistically significant.

- The environments are "empty squares" without features such as obstacles, or dead-ends, characteristic of real-world environments. These sorts of environments might be a very interesting benchmark for the long-term planning capabilities of the proposed method. I suppose there is nothing preventing using the PRM method for such environments as well?

# Minor comments
- Section 2: there are several misplaced/misnamed citations here. Please check and revise. Examples: Line 73: [9] refers to Hollinger et al., but in main text referred to as Karaman et al. - Line 76: Hitz et al. [5] does not match the corresponding item in bibliography
- Line 123: there is something missing here from the definition of $\mu_{th}$, the preceding sentence ends abruptly.
- Lines 185-186: please use consistent notation for $W^K$, $W^V$
- Table 2 heading: "WTIHOUT" --> "WITHOUT".
- Figure and table captions use bold face somewhat arbitrarily, a consistent style would be preferable.

**Summary Of Recommendation:**

This paper proposes an interesting new view of informative path planning as a reinforcement learning problem, and proposes a practical method for taking advantage of the reformulation, with good empirical performance. There are some concerns about generalization capability and applicability beyond simple topologies or well-known target processes. The authors have addressed these concerns and my comments in their rebuttal, and I recommend accepting the paper.

---

> ### Author Response · Authors · 2022-08-22
> **Reply to Reviewer HgVR**
>
> Thanks a lot for your positive feedback and compliments! We hope our answers below can resolve your concerns about CAtNIPP, and will be happy to respond to any further questions/comments you may have.
> ### Weaknesses 1 and 2:
> - Please also refer to our response on generalizability to Reviewer QZo2’s “weakness 1 and issue 1” above, where we clarify (and mitigate) the need for ground truth information, as well as discuss matters related to the generalizability and need for retraining of our approach in practice (especially towards real-robot deployments).
> Note that, despite these arguments, we also revised our manuscript to state that “once trained, CAtNIPP outperforms baselines in terms of the computing time” to make our argument fairer.
> ### Weakness 3:
> - The plus/minus values in Table 1 indicate the standard deviation (which we have clarified in the revised manuscript). We additionally added a new box plot figure showing the results of our comparison analysis (see attached file).
> - Regarding the concern with statistical significance, we thank the reviewer for bringing up this important issue! We performed paired t-tests (with Bonferroni correction) between the results of our approach (greedy variant) and each of the baselines, and reported the p-values here (as well as in the revised manuscript): $p=1.05 \times 10^{-4}$ for greedy-CMAES, and $p=4.96 \times 10^{-7}$ for greedy-RAOR, $p=3.33 \times 10^{-14}$ for greedy-RIG, $p=1.45 \times 10^{-7}$ for ts-CMAES, $p=3.63 \times 10^{-12}$ for ts-RAOR, and $p=2.33 \times 10^{-17}$ for ts-RIG . Using a Bonferroni correction for these multiple tests, we find that the final significance threshold is $p = 1.67 \times 10^{-3}$, for a standard, original $p < 0.01$ significance threshold. These t-test results indicate that the differences in performance between CAtNIPP and each of the baselines is most likely statistically significant.
> ### Weakness 4:
> - We would like to mention that our recent ongoing work has been focusing on autonomous exploration in obstacle-dense indoor environments, where only minimal changes to CAtNIPP were necessary to already obtain very interesting preliminary results. Please also refer to our reply to Reviewer 6v6E “Weakness 1 and Issue 1” (last paragraph).
> ### Issue 1:
> - We thank the reviewer for indicating these mistakes in the paper. We have revised the manuscript accordingly.

---

> > ### Comment · Reviewer_HgVR · 2022-08-25
> > **Thanks for responding!**
> >
> > Thank you for responding to my review. I am happy with the response and will maintain my positive rating for the paper. The results of the t-test along with the mentioned other mentioned corrections will be good to include in the final paper. The generalization results for now are sufficient and seem promising, and I hope that in future work the method can also be demonstrated in more convincing robotic applications.

---

### Official Review · Reviewer_6v6E · 2022-07-26

**Originality:** Very Good
**Technical Quality:** Very Good
**Clarity Of Presentation:** Very Good
**Impact:** 3

**Recommendation:**

Weak Accept: I recommend accepting the paper, but will not argue for my recommendation if the majority of other reviewers have a different opinion.

**Summary:**

This paper presents a deep reinforcement learning approach (CAtNIPP) for informative path planning (IPP), where the goal is to gain information by navigating a new environment while satisfying some resource budget. Notably, CAtNIPP appears to be the first RL based adaptive IPP solver. The proposed approach outperforms prior non-learning based IPP solvers both in terms of information gain and computing time on a set of randomly generated 2D navigation problems. CAtNIPP is also evaluated on a simple 2D navigation task on a physical robotic platform (TurtleBot3).

**Issues:**

(1) Simulation and physical experiments are so simple that the relevance of CAtNIPP to practical robotic problems is not clearly demonstrated. Adding more complex robotic control domains (or at least a more complex task specification) would significantly strengthen the contribution of the paper for CoRL.

**Quality Of The Limitations Section:**

Limitations are addressed clearly

**Reviewer Expertise:**

3: The reviewer is fairly confident that the evaluation is correct

**Robotics Focus:**

Sufficient demonstration on hardware

**Strengths And Weaknesses:**

Strengths:

(1) CAtNIPP is very clearly described and represents a novel application of recent advances in machine learning, notably RL and attention mechanisms, to an important problem in robotics (information path planning).

(2) CAtNIPP is very clearly positioned with respective to prior work. Namely, prior RL based IPP approaches exist, but only work for non-adaptive settings, while prior conventional IPP solvers are shown to achieve worse solution quality in a higher amount of time than CAtNIPP.

(3) The simulation experiments are very well designed and reported, with comparisons to relevant baselines and clear demonstration that CAtNIPP achieves generally higher solution quality in general than conventional approaches while being significantly faster (at least the greedy variant is).

Weaknesses:

(1) The main weakness of this work is that the experimental evaluation is very simplistic, both in simulation and in physical experiments. While the experiments do illustrate the benefit of CAtNIPP over prior approaches, this benefit is restricted solely to simple 2D navigation tasks in free space. Tasks with greater complexity involving realistic robot motion constraints and more complex specifications (eg. trying to gain information about different rooms in a house from image observations) would make the utility of CAtNIPP for practical robotic problems much more clear.

**Summary Of Recommendation:**

CAtNIPP is very well motivated and well described, and the simulation experiments do thoroughly evaluate and demonstrate the benefits of CAtNIPP with respect to prior IPP solvers. However, the main uncertainty I have with this paper is that the experiments are so simple that the relevance of CAtNIPP to practical robotic problems is not clearly demonstrated. I am still leaning towards an accept decision at the moment, but would be more confident in this decision if a stronger case can be made regarding the relevance of this paper for practical robotic problems.

---

> ### Author Response · Authors · 2022-08-23
> **Reply to Reviewer 6v6E**
>
> Thanks a lot for your helpful feedback! We are happy to hear that our ideas are novel and clear to you. We hope our answers below can resolve your concerns about CAtNIPP, and will be happy to respond to any further questions/comments you may have.
> ### Weakness 1 and Issue 1:
> - Although CAtNIPP is indeed trained and tested on Gaussian distribution-based simulations (in addition to our robot experiments), we do not believe that these simulations are necessarily (too) oversimplified for a few reasons. First, previous IPP planners such as Hitz et al. and Popovic et al. have been used for field experiments for cyanobacteria abundance and agriculture monitoring tasks, while modeling the environment as a GP similar to our work. Due to hardware limitations, we could only conduct an illumination-based task for now, but we expect and plan to apply CAtNIPP to more realistic-world tasks in the future.
> - Second, and as discussed in our response to Reviewer QZo2’s “weakness 1 and issue 1” above, we believe that such a high level representation actually allows our CAtNIPP agent to learn a high-level strategy for IPP, which reasons about the relationship between motion (i.e., neighbor selection) and short-/long-term metrics such as exploration and exploitation. This strategy helps with generalizability, as evidenced by the additional results on never-seen-at-training information maps reported in our supplementary material. More importantly, we envision that our simplified PRM/GP model allows us to position RL at the appropriate level of control/planning, to really ensure we draw out the main advantages of learning (e.g., balance of short- and long-term objectives, discretized space for action selection/planning, development and use of a reactive, stochastic policy). Lower-level details (e.g., specific motion/motor control of the robot to reach nodes of the PRM) can then be handled by existing, conventional methods where RL could likely not introduce significant improvements, thus simplifying implementation on hardware without really limiting the range of applications that CAtNIPP could be used for. This observation and philosophy is aligned with many recent learning-based methods for complex robotic control/planning tasks, such as multi-agent pathfinding, where state-of-the-art methods still focus on abstracted, gridworld-like problems, where strategies can be learned that then can still be adapted/translated to work on more realistic scenarios.
> - Finally, and admittedly out of the IPP scope of this paper, we would like to mention that our recent ongoing work has been focusing on autonomous exploration in obstacle-dense indoor environments, where only minimal changes to CAtNIPP were necessary to already obtain very interesting preliminary results. Specifically, we needed to adapt the inputs of the model to better suit the general assumption of exploration tasks, and how it is used by the agent to both reason about information gathering and now plan efficient paths in the presence of obstacles (in an unknown environment, whose map needs to be constructed first). There again, training on simplified environments is crucial to speed up training and allow the agent to focus on learning a high-level, more generalizable strategy, which can then be translated back to robot commands by low-level conventional controllers.

---

> > ### Comment · Reviewer_6v6E · 2022-08-24
> > **Thank You for Your Response**
> >
> > I appreciate that prior IPP planners were also evaluated on similarly simple tasks, so the scientific contribution of this work is still strong. My only reservation is that for CoRL, typically one would expect to see evaluation on more interesting robotic tasks, so my only concern is regarding the alignment of CAtNIPP with the goals of the CoRL conference. In any case, I still believe that this is a strong paper, and my evaluation of this work remains positive (Weak Accept).

---

### Official Review · Reviewer_8gWj · 2022-07-26

**Originality:** Good
**Technical Quality:** Good
**Clarity Of Presentation:** Very Good
**Impact:** 3

**Recommendation:**

Weak Accept: I recommend accepting the paper, but will not argue for my recommendation if the majority of other reviewers have a different opinion.

**Summary:**

The paper studies the informative path planning problem (IPP) in a Gaussian random process/field (GP). The objective is to maximize the variance reduction at points where the field values exceed the GP upper confidence bound, while respecting the path length constraints. The sets determined by the upper confidence bound change with each additional measurement motivating an online adaptive algorithm.

The main contribution of the paper is an algorithm based on deep reinforcement learning (DRL) to compute approximate solutions to the IPP problem. An extension of the DRL algorithm based on receding horizon planning is also provided which provides higher quality solutions. In simulations, the proposed algorithms (DRL and DRL+receding horizon) yield more informative paths when compared to previously proposed algorithms. In addition, the computation time of DRL (excluding training time) is lower. The idea to use reinforcement learning for IPP is quite interesting!

**Issues:**

1. Since the generalization capability is limited, I am not convinced that it is correct to say the proposed approach outperforms baselines in terms of compute time. It would be more appropriate to mention the conditions under which the proposed approach is preferable over the baselines i.e., environments with fixed parameters.

2. If the experiments section only considers functions that are samples from a GP, it would be more thorough if other types of functions are considered as well. For example, piecewise constant, Holder smooth functions, etc.

3. The discussion on the completeness of the algorithm should be included.

4. The paper mentions the metric used for comparison is the uncertainty remaining after finishing the mission. However, it is not clear whether this is over all n' locations where the quantity is to be inferred or if it is over the upper confidence bound sets determined at the end of each algorithm.

**Quality Of The Limitations Section:**

Additional details required

**Reviewer Expertise:**

4: The reviewer is confident but not absolutely certain that the evaluation is correct

**Robotics Focus:**

Relevant but unlikely to deploy to hardware in near future

**Strengths And Weaknesses:**

Strengths:

The biggest strength of the paper lies in the formulation of the IPP problem as a reinforcement learning problem. The parts of the formulation that I found interesting are discussed below:

1. Construction of the state/observation
       At each step of an online adaptive algorithm for IPP, the robot needs to make a decision on which node to visit next based on the information collected. The paper constructs a state vector which encodes the current position of the robot, the state of the environment (as predicted by the GP), and the remaining budget (as a binary mask).

2. Action/Control space:
       Since the robot is operating on a graph, its action space (which node to visit next?) is determined by the degree of its current vertex. However, most RL methods have fixed-dimensional action spaces. The paper circumvents this issue by using Pointer Networks, which allows the algorithm output to depend on the degree of the current vertex. This is a neat idea!


Weaknesses:

1. Since the output of the proposed approach is a stochastic policy, it is not clear whether the algorithm will always reach the goal/destination vertex. If this is not guaranteed, then the algorithm does not always provide a feasible solution.

2. While the proposed approach works very well in simulation, the limited generalizability is concerning. The authors mention that the model requires retraining if parameters, such as sensor type or task specification, change. However, the issue could extend to other parameter changes. For example, environment dimensions, kernel hyperparameters, change of kernel, connection factor in the PRM, number of nodes in the PRM, etc. Since the computational resources required for training is significant (24 hours training time, distributed training on 4 GPUs), the generalizability is a serious question since the other baselines do not require retraining.

3. The experimental section is quite extensive (10 trials on 30 instances for each budget). However, there are a few missing details:
   - What types of functions are considered? It would be interesting to see how the policy works on functions that are not samples from a GP.
   - What is the environment size considered? Is it the same considered for training the policy i.e., the unit square?
   - What is the source of randomness between the experiments? Is it the probabilistic roadmap that changes between trials while keeping the number of nodes fixed?

4. The chosen architecture has yielded good results for the IPP problem. However, it would be nice to get more insight into the design process:
   a. Is the selection of \mu=0.4, \beta=1 fixed for all environment types? What if the GP has non-zero mean and a priori variance > 1?
   b. It seems that the attention module is crucial to the network construction. Does the same hold for the other modules of the architecture? For example, is the LSTM module required for the decoder?

**Summary Of Recommendation:**

The paper proposes a novel approach to solve the IPP problem using deep reinforcement learning. While the generalization/applicability of the algorithm to environments with different parameters is limited, the idea is quite interesting. With some more experiments and clarifications, I propose a weak accept.

---

> ### Author Response · Authors · 2022-08-22
> **Reply to Reviewer 8gWj**
>
> Thank you very much for your time and valuable feedback, we are happy to hear that you found our paper and RL formulation interesting!  We hope our answers below can resolve your concerns about CAtNIPP, and will be happy to respond to any further questions/comments you may have.
> ### Weakness 1 and Issue 3:
> - We guarantee the completeness of CAtNIPP (i.e., reaching the destination) by applying a budget mask to explicitly filter out actions that would lead to over-budgeted trajectories. That is, in practice, the agent is forced to reach the destination before it runs out of budget. The details about the budget mask and its use to enforce completeness can be found on lines 152-157, and we have further clarified this important feature in our revised manuscript.
> ### Weakness 2 and Issue 1:
> - We thank the reviewer for bringing up these generalizability issues. CAtNIPP has already been shown to generalize to arbitrary numbers of nodes and connection degrees of the underlying PRM, as reported in the supplemental materials submitted alongside our original manuscript. These additional results also consider general ground truth information maps, never seen at training time. We have discussed this central point mentioned by several reviewers in detail, in our answer to Reviewer QZo2’s “weakness 1 and issue 1” above.
> We revised the manuscript to state that “once trained, CAtNIPP outperforms baselines in terms of the computing time” to make our argument fairer and clearer.
> ### Weakness 3 and Issue 2:
> > What types of functions are considered?
> - In the experiments section, we consider scenarios where the true information map is a Gaussian-mixture function (built from randomness parameters identically to the training ones, but not exactly the same as the training maps). In the supplemental material submitted with the paper, we considered both Gaussian (drastically different from the training ones) and non-Gaussian (including square-like distribution) mixtures.
>
> > What is the environment size considered?
> - It is the unit square.
>
> > What is the source of randomness between the experiments?
> - The randomness mainly comes from the PRM. For trajectory sampling variants, additional randomness comes from the way in which we sample actions from the stochastic policy.
> ### Weakness 4:
> - $\mu=0.4$, $\beta=1$ are fixed for all environments, and we select such parameters following Hitz et al., for consistency.
> We believe that starting with a GP with non-zero mean and a priori high variance will not be particularly problematic for CAtNIPP, since such a case may be similar to intermediate steps during a standard episode. We uploaded some gifs to show that planned trajectories are still rational when starting with an accurate but incomplete initial GP (i.e., the peaks shown are correct, but some are missing), with high initial uncertainty. We also show a few cases where the initial prior is non-zero, and is not aligned with the ground truth (i.e., inaccurate priori), also with initially high variance. While these two cases currently fall out of scope of our paper (as this is not the standard formulation of most IPP problems), these gifs show CAtNIPP still exhibiting reasonable performance without any retraining. Our model naturally seems to perform better in the presence of an incomplete prior, as the (limited) information present is at least trustworthy, while inaccurate priors seem to throw the agent off initially, i.e., until their belief is “corrected” by enough measurements. We are very interested in looking at such cases in more depth, where an initially incomplete/inaccurate prior has to be simultaneously leveraged and corrected by the agent as it gets new measurements, and thank the reviewer for this suggestion! However, we envision that this new line of research will likely need more fundamental changes to our approach (to truly yield near-optimal results, and not just reasonable performance), and as such is left as future work.
> - Regarding our use of an LSTM cell, we experimentally found that removing this cell does not significantly impact performance (less than 10%). The final covariance trace of our greedy variant without LSTM in the same evaluation tests is: 22.73, 7.52, 4.11, and 2.93 for budget 6, 8, 10, and 12 respectively (in comparison, the full CAtNIPP model with LSTM results in 22.86, 7.72, 3.97, and 2.70 final trace).
> ### Issue 4:
> - We are sorry that we didn’t make it clear in the paper, we evaluate the uncertainty over the upper confidence bound sets. We have clarified this in the updated manuscript.

---

> > ### Author Response · Authors · 2022-08-22
> > **Example GIFs of CAtNIPP with incomplete and inaccurate priors**
> >
> > **Comment:**
> >
> > Example GIFs of CAtNIPP with incomplete and inaccurate priors
> >
> > **Zip File:**
> >
> > /attachment/ccb8d8484f3f9a21466d252e8edf50c117e7610a.zip

---

> > > ### Comment · Reviewer_8gWj · 2022-08-24
> > > **Thank you and response to reply**
> > >
> > > Thank you for the detailed response to the initial review! The comments have improved my understanding of the paper and have addressed the weaknesses/issues raised.
> > >
> > > I have one comment regarding the evaluation metric (Issue 4). In the response, the authors mentioned the uncertainty is evaluated over the upper confidence bound sets for each algorithm. As per my understanding, this set is determined by the chosen measurement locations by each algorithm (since it involves the predicted mean/variance of the GP model). For example, consider two algorithms $A$ and $B$ and their corresponding UCB sets $X_A$, $X_B$ at the end of their computation. It might be that $X_A \neq X_B$. As a result, the final uncertainty is measured across different sets for different algorithms. This makes it difficult to judge the performance of the algorithms. Would it be better to measure the uncertainty across all locations of interest i.e., the set $X^*$ of $n'$ locations?

---

> > > > ### Author Response · Authors · 2022-08-25
> > > > **Response to the evaluation metric**
> > > >
> > > > Happy to hear that our response solves your concern!
> > > >
> > > > In theory, a case where $X_A \neq X_B$ could indeed happen, thus potentially affecting the accuracy of comparison between these two algorithms. However, we empirically observe from our results that the final UCB set is very close to $\overline{X}$ for all algorithms considered, since they all efficiently cover the entire domain to build a high-quality belief (as evidenced by the very low final KL divergence between belief and ground truth). Therefore, we believe that relying on each algorithm's UCB still yields an accurate comparison of their performances. We note that this choice is also consistent with previous work (Hitz et al. and Popovic et al.).

---

### Official Review · Reviewer_QZo2 · 2022-07-28

**Originality:** Fair
**Technical Quality:** Good
**Clarity Of Presentation:** Excellent
**Impact:** 4

**Recommendation:**

Weak Accept: I recommend accepting the paper, but will not argue for my recommendation if the majority of other reviewers have a different opinion.

**Summary:**

This paper introduces a deep reinforcement learning (dRL) method for solving the adaptive IPP problem. The proposed dRL architecture leverages a self-attention layer to inform local decisions with global information, allowing for non-myopic decision making. The authors provide thorough experimental and simulated demonstrations of strengths of their architecture and comparisons to state of the art IPP solutions.

**Issues:**

- Strong justification of using RL for the IPP problem is required: especially in the proposed approach where training requires ground truth to be available
- Authors could benefit from emphasising the relatively small improvement gained from trajectory sampling rather than employing the greedy selection method: it would emphasise the naturally non-myopic nature of their policy. Other IPP solutions only embed non-myopic behaviour from trajectory sampling.
- Caption of Table 2 has typo: should say “WITHOUT”
- In subsection: Adaptive Replanning, authors should make it clear that the mean and variance in the UCB is prior to fusing measurements along the future trajectory.
- Authors should justify why the negative correction reward proposed solves the discrepancy between training and IPP objectives.


**Quality Of The Limitations Section:**

Limitations are addressed clearly

**Reviewer Expertise:**

5: The reviewer is absolutely certain that the evaluation is correct and very familiar with the relevant literature

**Robotics Focus:**

Sufficient demonstration on hardware

**Strengths And Weaknesses:**

Strengths:

- Well written.
- Well placed in existing literature
- Provides a novel solution to a well researched problem.
- Proposed solution has benefits outside of only solution quality/efficiency (provides attention map)
- Supplementary videos and gifs are good
- Experimental validation is strong, particularly in demonstrating the benefit of the attention layer.

Weaknesses:

Certain technical details that should be emphasised are skimmed over, for example:
- As with any learning method, issues arise with generalisability. WIth a training time of 24 hours with a substantially powerful workstation, - This seems like a major limitation of the proposed method:
    - e.g. what if the true function of interest is not well represented by the learned kernel? Retraining would be required.
    - During training, the ground truth function of interest must be available. If for example the function of interest in Fig. 4 were inverted (low interest where high interest areas once were), the attention layer learned would be obsolete. Retraining would be required.
- Training reward is not equivalent to the IPP objective, and it is unclear how the negative correction reward proposed recovers equality.

This approach requires prior knowledge of the ground truth function to learn a good policy, which diminishes its utility for field-based/realistic scenarios greatly. If ground truth is available for training, no policy needs to be learned at all.

**Summary Of Recommendation:**

Weak accept, provided the authors add significant justification for their method given the limitations regarding generalisation/requirement for ground truth during training. More emphasis should be placed on the proposal and benefits of the attention layer, as this is (in my opinion) the main strength of the paper.

---

> ### Author Response · Authors · 2022-08-22
> **Reply to Reviewer QZo2**
>
> Thank you very much for your positive feedback on the novelty, written work, and experiments of our paper! We hope our answers below can resolve your concerns about CAtNIPP, and will be happy to respond to any further questions/comments you may have.
> ### Weakness 1 and Issue 1:
> - From prior literature, as well as our experience, the main challenge in adaptive IPP is how to effectively utilize the robot’s belief to make/sequence non-myopic movement decisions. Most conventional state-of-the-art methods address this challenge by relying on sampling-based optimization of a long trajectory, which is computationally expensive and rarely achieves optimal solutions (even with a meta-heuristic strategy). In this context, our initial hypothesis was that RL should be able to allow the agent to learn the underlying dependencies within the robot’s belief over the full domain, which can then be used by the agent to build efficient trajectories sequentially, thus avoiding the need for (expensive) optimization. Our results support this hypothesis, showing that our RL-based planner efficiently learns to utilize its entire belief to make better decisions than conventional methods, while drastically cutting down on computing times. Therefore, we believe that our work and results can open the way and encourage more research in developing RL-based approaches for IPP. We have revised the manuscript to clarify this.
> - As described in the limitation section, our proposed method is sensitive to changes in the hyperparameters of (1) the GP’s kernel function and (2) the upper bound confidence (UCB) that defines high-interest areas. Currently, CAtNIPP would require retraining if any of these parameters were changed significantly. In practice, the hyperparameters of the GP’s kernel are decided by key characteristics of the actual sensor equipped on the robot (e.g., range or resolution of a camera). In particular, the size of the GP kernel is related to the ratio between the dimensions of the search domain and the sensor coverage area. When deploying CAtNIPP on-robot, we assume that the sensor of a robot will likely not be frequently changed. The hyperparameters of the UCB are decided by the (human-defined) specifications of the task at hand, which can vary across tasks. However, based on our experience, for a given type of real-world task (e.g., search and rescue, fire detection, agriculture monitoring),  the definition of “high-interest” is usually rather static. Therefore, we would expect that, after an initial training based on these (GP and UCB) parameters, our learned model will likely not need any further training in practice, since retraining would only be needed if the task or onboard sensor are drastically changed. We have revised the manuscript to clarify this.
> - Second, we would like to clarify that, although a ground truth is indeed required for training, the true properties of information maps the learned model will be used/deployed in does not need to be known. That is, our preliminary generalization results (included in our supplemental material) suggest that CAtNIPP allows the agent to learn a general strategy for IPP, instead of overfitting to one specific class of ground truths. That is, we showed that our model can be trained on randomly generated ground truths, and still generalizes to completely different, never-seen-at-training environments, such as different Gaussian mixture models, or even non-Gaussian, handcrafted distributions, while still exhibiting better performances than the baselines. We have revised the manuscript to clarify this.
> Our ongoing and future work will still focus on improving generalization, likely by first adding these key (GP and UCB) hyperparameters as inputs of the model. We believe that this may allow us to develop a new model that may not even need retraining if robot or task properties are changed. We will revise the manuscript to highlight and clarify the key issues and properties of CAtNIPP with respect to generalizability that we discussed above.
> - We apologize for our over-stated description of the training time (and necessary hardware) in the original manuscript.  Although our tested model was actually trained on a quad-GPU machine for 24 hours, we now know that such resources are not necessary to fully train a CAtNIPP model. Since the submission of the original manuscript, we have tried to train the same model on a machine with a single NVIDIA 2080 Super GPU, with a batch size of 256. There, our newest results show that training for about 12 hours already allows us to reach the same level of performance. This is explained by the fact that we historically trained our model for extended periods of time (24h or more) to ensure it fully converged, but this amount of training is not necessary and convergence can be achieved earlier without any effect on final performance. We have clarified this in the revised manuscript.

---

> > ### Comment · Reviewer_QZo2 · 2022-08-27
> > **Thank you for clarifying**
> >
> > Thank you for the detailed response, this clarifies many of the issues raised. Regarding generalisation experiments, these should be moved into the main document as they are an important factor when considering the applicability of your method.

---

> ### Author Response · Authors · 2022-08-22
> **Reply to Reviewer QZo2 (Part 2)**
>
> ### Issue 2:
> - We thank the reviewer for this very good suggestion. After submitting the paper, we actually presented and discussed this work with some of our collaborators, and unanimously obtained the same feedback! We agree that the greedy variant of CAtNIPP only sacrifices very little performance, for huge improvements in computing time, likely making it the most appealing version of our model. That being said, we still would like to highlight that our ts variants help output more robust solutions in practice, by further refining the performance of learned policies in extreme cases (where the greedy ones sometimes do poorly). We have revised the manuscript to emphasize the greedy variants, while still mentioning trajectory sampling ones, as you recommended.
> ### Issue 3:
> - We thank the reviewer for indicating the error, which we have fixed in the revised manuscript.
>
> ### Issue 4:
> - We thank the reviewer for indicating this issue. We have clarified all the uses of the UCB and of the GP for future predictions throughout the revised manuscript, to really make sure the reader understands which version of the GP is used each time (before/after measurement).
>
> ### Weakness 2 and Issue 5:
> - We apologize for the lack of clarity surrounding this point in the original manuscript. The negative correction reward proposed does not solve the discrepancy between training and IPP objectives, but aims at alleviating it (which we empirically observed stabilizes training, while still exhibiting high performances). That is, while the true IPP objective would only have the reward be the difference between the covariance trace of the GP before and after each measurement, we simply proposed to renormalize this difference to help stabilize training. If the normalization term were constant, and $gamma=1$, then this would simplify back to the IPP objective ($Return=r_1+r_2+...+r_n=(Tr(P^0)-Tr(P^1))+(Tr(P^1)-Tr(P^2))+ … + (Tr(P^{n-1})-Tr(P^n)))=Tr(P^0)-Tr(P^n)$), but a constant normalization term cannot be used that works for all cases (see in Figure 3 how the trace is decreasing drastically during early stages of the episode, but much slowly at later stages). Therefore, we opted for a dynamic normalization term, which introduces a small bias that our correction helps reduce, but not entirely remove. However, we note that our results support the fact that this slight bias does not seem to impact final performances, as our models are still obtaining covariance trace reductions that outperform SOTA IPP baselines. In other words, optimizing our slightly modified reward function still allows the agent to learn the right strategy for the final, true IPP objective used at testing, indicating that this mismatch is most likely negligible.

---

> > ### Author Response · Authors · 2022-08-22
> > **Training comparison between the dynamic normalization term and constant normalization term**
> >
> > Training comparison between the dynamic normalization term and constant normalization term

---

### Comment · Area_Chair_GBYc · 2022-08-19
**Meta Review Comments**

Thank you authors and reviewers, here is a short summary of some of the key strengths and weaknesses identified by the reviewers.
Authors & reviewers, please engage in a discussion regarding the issues raised by the individual reviews.

Strengths:
- a well written work
- benefits of attention layer is shown clearly in presented experiments
- a novel solution with interesting approach where RL and attention mechanisms are applied to IPP

Weaknesses:
- questions around generalization and retraining are raised by reviewers QZo2, 8gWj
- benefits of the approach are only evaluated on relatively simple environments, raising some concern regarding applicability to real world
  robotics problems
- more detailed elaboration to what extent negative correction reward is sufficient to ensure equivalence to the IPP objective may be required

---

> ### Author Response · Authors · 2022-08-22
> **Reply to Area Chair GBYc**
>
> Thank you very much for your time and help! Our answers to the key weaknesses and points of improvement in our manuscript can be found in our detailed answers to the reviewers:
> ### Weakness 1
> We have discussed this central point mentioned by several reviewers in detail, in our answer to Reviewer QZo2’s “weakness 1 and issue 1”.
> ### Weakness 2
> We have discussed this issue in our answer to Reviewer 6v6E’s “weakness 1 and issue 1”.
> ### Weakness 3
> We have discussed this issue in our answer to Reviewer QZo2’s “weakness 2 and issue 5”.

---

### Meta-Review · Area_Chair_GBYc · 2022-09-05

**Recommendation:** Accept (Poster)
**Confidence:** 3

**Metareview:**

The 4 reviewers all recommended a weak accept rating for this work. The work makes a relevant contribution in the area of informative path planning.
Overall some of the key strengths and weaknesses include:

Strengths:

-  a well written work that is easy to follow
-  benefits of attention-based neural network is shown in experiments
-  a novel solution with interesting approach where RL and attention mechanisms are applied to IPP

Weaknesses:

- questions around generalization and retraining are raised by reviewers QZo2, 8gWj and not fully resolved
- benefits of the approach are only evaluated on relatively simple environments, expensive compute requirements
- negative correction reward serves only as a heuristic to approximate IPP objective

A concerning weakness of the approach (shared by several IPP algorithms) is the limited complexity of the presented robotic experiments, featuring simple 2D navigation environments. This may be a concern for the CoRL audience and may limit the significance of the proposed approach.

I follow the four reviewers' assessment and agree with the "weak accept" rating.